# The effectiveness of botulinum toxin for temporomandibular disorders: A systematic review and meta-analysis

Ravinder S. Saini[1], Muhammad Ali Abdullah Almoyad[2], Rayan Ibrahim H. Binduhayyim[1], Syed Altafuddin Quadri[1], Vishwanath Gurumurthy[1], Shashit Shetty Bavabeedu[3], Mohammed Saheer Kuruniyan[1], Punnoth Poonkuzhi Naseef[4], Seyed Ali Mosaddad[5,6]*, Artak Heboyan[5,7,8]*

1 Department of Dental Technology, COAMS, King Khalid University, Abha, Saudi Arabia, 2 Department of Basic Medical Sciences, COAMS, King Khalid University, Abha, Saudi Arabia, 3 Department of Restorative Dental Sciences, College of Dentistry, King Khalid University, Abha, Saudi Arabia, 4 Department of Pharmaceutics, Moulana College of Pharmacy, Perinthalmanna, Kerala, India, 5 Department of Research Analytics, Saveetha Dental College and Hospitals, Saveetha Institute of Medical and Technical Sciences, Saveetha University, Chennai, India, 6 Student Research Committee, School of Dentistry, Shiraz University of Medical Sciences, Shiraz, Iran, 7 Department of Prosthodontics, Faculty of Stomatology, Yerevan State Medical University after Mkhitar Heratsi, Yerevan, Armenia, 8 Department of Prosthodontics, School of Dentistry, Tehran University of Medical Sciences, Tehran, Iran

* mosaddad.sa@gmail.com (SAM); heboyan.artak@gmail.com (AH)

**Data Availability Statement:** All relevant data are within the manuscript and its Supporting Information files.

## Abstract

### Objective

The current body of research on utilizing botulinum toxin (BTX) to manage temporomandibular disorders (TMDs) has not yet yielded definitive conclusions. The primary objective of this study was to determine the effectiveness of BTX in pain reduction for TMDs compared to placebo and other treatments. The secondary outcomes evaluated were adverse events, maximum mouth opening, bruxism events, and maximum occlusal force.

### Materials and methods

A literature search was performed on PubMed, Dimension Publication, Scopus, and Google Scholar. The RoB 2 tool was used for quality assessment. The mean differences in pain scores were estimated to measure the effect of BTX on pain reduction. For adverse events, the risk ratio for the incidence of side effects was calculated.

### Results

Two hundred and sixty non-duplicate articles were identified; however, only 14 RCTS were included in this review. The total study population included 395 patients. The overall risk of bias showed a low to moderate quality of evidence. Results from 6 studies were reported only narratively; four studies were used for meta-analysis on pain reduction, and five were used for meta-analysis on adverse events. The control used in the meta-analysis was placebo injections. Results of the meta-analysis for pain reduction were statistically insignificant for the BTX group with mean differences at MD = −1.71 (95% CI, −2.87 to −0.5) at one month, -1.53 (95% CI, −2.80 to −0.27) at three months, and -1.33 (95% CI, −2.74 to 0.77) at

**Funding:** Deanship of Scientific Research at King Khalid University partially funded this work through Large Group RGP 2/348/44.

**Competing interests:** The authors declare no conflict of interest.

six months. This showed that BTX treatment was not significantly better than placebo for a reduction in pain scores at 1, 3, and 6 months. Regarding safety, the placebo group showed a relative risk of 1.34 (95%CI, 0.48–6.78) and 1.17 (95%CI, 0.54–3.88) at 1 and 3 months respectively. However, the risks were not statistically significant. There was also no difference in the effectiveness of BTX compared to placebo and other treatments for maximum mouth opening, bruxism events, and maximum occlusal force.

## Conclusion

BTX was not associated with better outcomes in terms of pain reduction, adverse events, maximum mouth opening, bruxism events, and maximum occlusal force. More high-quality RCTs are needed to better understand this topic.

## Introduction

The scientific term temporomandibular disorders (TMD) encompasses debilitating illnesses that impact the temporomandibular joint (TMJ), muscles involved in chewing, and related anatomical structures [1, 2]. Patients frequently exhibit symptoms such as discomfort during jaw movement, neck pain, limited jaw movement, joint noises (crepitus), difficulty in mouth opening (trismus), ringing in the ears (tinnitus), ear pain, pain around the eyes (periorbital pain), and headaches [3, 4]. Various risk factors have been identified as potential contributors to TMDs. These include trauma, dental misalignment, joint hypermobility, and anatomical, psychological, and systemic diseases [5–7].

Al-Jundi et al. [8] reported that 44% of the population has this illness, but only 25% seek expert help. The study found that the prevalence of 'treatment need for TMD' in adults was approximately 15–16%. TMD treatments range from non-pharmacologically conservative to invasive surgery.

The first-line treatment for TMDs includes physiotherapy, behavioral therapy, occlusal splints, and lasers [9–11]. TMJ arthrocentesis has also emerged as an effective procedure and alternative to surgical intervention [12, 13]. The upper joint space is cleaned using Ringer's lactate to remove inflammatory mediators in this procedure. Various therapeutic agents, such as Ringer's lactate, steroids, platelet-rich plasma (PRP), and hyaluronic acid (HA), have been successfully injected into the TMJ space to alleviate pain and improve quality [9, 12–16].

Hyaluronic acid (HA) is a linear polysaccharide naturally present in the synovium, vitreous humor, and connective tissue [17, 18]. It is used because of its anti-inflammatory and lubricating properties. It also reduces mechanical wear, supports the cartilage tissue repair process, and stimulates the production of endogenous acids by synovial cells [17]. It was found to significantly decrease pain after 12 months in a study by BJØRNLAND et al. [19]. Better results for pain relief were found for five weekly injections than for a single injection [20]. In the literature, HA injections have shown greater efficacy in pain reduction compared to corticosteroids (CS) and lower or similar effectiveness when compared to platelet-rich plasma (PRP) [12, 20–24]. Ozone therapy is another technique used to manage TMD. In a systematic review by Argueta-Figueroa et al. [25], ozone therapy was found to improve pain and maximal mouth opening in TMD patients.

Botulinum toxin (BTX) has become a popular treatment for TMD in recent years. BTX is a strong neurotoxin produced by the gram-negative, anaerobic, spore-forming bacterium Clostridium botulinum [6]. Since the late 1970s, botulinum toxin serotypes A and B (BTX-A and

BTX-B) have been used to treat neuromuscular diseases such as cervical dystonia, blepharospasm, and hemifacial spasm [26]. This neurotoxin irreversibly binds to presynaptic cholinergic junctions, resulting in diminished muscular activity. Because many TMDs are linked to clenching, bruxism, or parafunctional mandibular actions, muscle activity inhibition may help control them [2, 27, 28]. Furthermore, it has been shown that BTX can reduce central pain sensitization and chronic pain through its impact on pain neurotransmitters and inflammatory mediators, including glutamate, calcitonin gene-related peptide, and substance P [29–33].

Owing to its capacity to reduce muscle activity and alleviate pain, BTX has emerged as a promising therapeutic option for TMD, with existing clinical reviews endorsing its effectiveness in TMD treatment [34, 35]. Nevertheless, studies have reported conflicting findings regarding the effects of botulinum toxin type A (BTX-A) in managing painful TMDs. A Cochrane systematic review by Sidebottom et al. [36] found no conclusive evidence that BTX-A improves myofascial pain [36]. According to another analysis, there is a lack of unanimity regarding the effects of BTX-A application on masticatory muscles in individuals with bruxism despite its increasing utilization in dentistry [29]. This study aimed to determine whether botulinum toxin can treat TMJ problems. Its efficacy will also be compared to that of other treatments. This study records and analyzes adverse events to assess BTX safety in patients with TMD.

## Materials and methods

The Preferred Reporting Items for Systematic Reviews and Meta-Analyses (PRISMA) standards were used to report the findings of this systematic review and meta-analysis, which was performed following the Cochrane Handbook for Systematic Reviews of Interventions [37]. The protocol used for this systematic review was the registered International Platform of Registered Systematic Review and Meta-Analysis Protocols (INPLASY) (2023110101).

### Research questions and study objective

The research questions for this review were developed using the PICO framework. Population: patients with temporomandibular disorders diagnosed through the Diagnostic Criteria for Temporomandibular Disorders (DC-TMD), the International Classification of Sleep Disorder (ICSD), or clinical examination by a clinical expert; intervention-injection of botulinum toxin, comparator-placebo, no treatment or treatment with active intervention; outcomes: pain, mouth opening, adverse events, etc.

I.  What is the effectiveness of botulinum toxin in pain reduction for temporomandibular disorders compared to placebo and other conventional treatments?

II.  What is the level of safety associated with the use of BTX in the treatment of TMJ disorders compared with placebo and other conventional treatments?

The study's primary outcome was pain reduction as measured using a 0–10 or 0–100 VAS scale. The secondary outcomes were adverse events, maximum mouth opening, bruxism events, and maximum occlusal force.

### Information sources and search strategy

A preliminary study on the topic was first conducted to identify the prominent and relevant keywords related to the topic. Keywords like "botulinum toxin," BTX-A, temporomandibular disorder, TMD, myofascial pain, bruxism, and their related terms were used to construct search strings used in PubMed, Dimension Publication, Scopus, and Google Scholar. The

**Table 1. Search string.**

| Database | Search String |
|---|---|
| PubMed | ("botulinum toxin" OR "Botulinum Toxins"[Mesh] OR botox OR BTX OR BTX-A) AND ("temporomandibular disorder" OR TMD OR "TMJ disorder" OR "temporomandibular joint" OR "Temporomandibular Joint"[Mesh] OR "Temporomandibular Joint Disorders"[Mesh] OR "myofascial pain" OR bruxism OR "Bruxism"[Mesh] OR "masticatory muscle") |
| Scopus | (orthodontic OR malocclusion) AND ("tooth movement" OR "dental arch") AND ("removable orthodontic appliance" OR "orthodontic bracket" OR elastic OR "dental aligner") |
| Google Scholar | ("botulinum toxin" OR botox OR BTX OR BTX-A) AND ("temporomandibular disorder" OR TMD OR "TMJ disorder" OR "temporomandibular joint" OR "myofascial pain" OR bruxism OR "masticatory muscle") |
| Dimension Publication | (orthodontic OR malocclusion) AND (tooth movement OR dental arch) AND (removable orthodontic appliance OR orthodontic bracket OR elastic |

The search in PubMed was filtered for only randomized controlled trials. No filter was applied to any other database. The database search results were then exported into the Zotero reference citation manager. Only the first ten pages of Google Scholar results were exported. Tools in the Zotero software were used to conduct a study selection process, following the eligibility criteria.

search strings comprised text words, mesh terms, and Boolean operators. The search strings used are listed in Table 1.

## Eligibility criteria

This review included only articles written in English and published from the database's inception until November 1, 2023. Only randomized controlled trials (RCTs) with parallel or crossover design and adult-only (≥18 years) populations were included. Clinical trials without an adult-only (≥18 years) population were eligible only if the results for the adult population were reported separately. The trial's primary objective was to examine the effectiveness of BTX in managing TMD. This objective is consistent with the PICO framework used in this review. Observational studies, systematic reviews, meta-analyses, comments, and concept papers were excluded from the analysis. Furthermore, articles that failed to provide information regarding the efficacy of BTX in treating patients with TMD were removed from the study.

## Risk of bias assessment

The Risk of Bias 2 (RoB 2) tool from the Cochrane Organization was used to appraise study quality [38]. The tool assesses five dimensions of bias: randomization, variations from intended interventions, missing outcome data, outcome assessment, and reported result selection. Robvis was used to create a risk of bias graph and a summary plot.

## Data extraction

After study selection was completed and eligible articles were identified, the following data were retrieved from each and presented in a predefined Excel spreadsheet: study authors, publication year, country of study, RCT Design, participant demographics, type of TMD, diagnostic method, intervention details, comparator details, outcomes assessed, and follow-up period.

## Data analysis

A meta-analysis was performed using Review Manager version 5.4. When calculating the effectiveness of BTX in pain reduction, a mean difference (MD) was used to pool the results of the VAS scales across studies. The Inverse Variance statistical method and the random effects

analysis model were used. To examine the safety of BTX treatment, the number of adverse events in each group was pooled, and a risk ratio (RR) was calculated. The Mantel-Haenszel statistical method and random effects analysis method were used. The $I^2$ statistic was adopted to assess study heterogeneity, and a p-value of 0.05 was adopted as the significance threshold.

## Results

### Study selection

The literature search yielded 272 articles: 61 from PubMed, 53 from Scopus, 58 from Dimensions Publication, and 100 from Google Scholar. Twelve duplicates were excluded. After title and abstract screening, 238 articles were removed because of topic irrelevance. The remaining 22 articles were subjected to full-text screening; eight were excluded for reasons shown in the PRISMA flowchart, and only 14 were included in this review. The PRISMA flowchart below illustrates the study selection process (Fig 1).

### Methodological quality assessment results

Overall, the RCTs were good quality evidence (Figs 2 and 3), with the most concerns being on the domain of 'bias in the selection of reported results.' No study was low risk in this domain, and all except Ondo et al. [48], which had high risk, had 'some concerns.' The other domains were, on average, low risk. This implies that the evidence in this review is low risk but has some methodological concerns.

### Results of data extraction

The data extraction results are shown in the appendix of the study descriptor table (Table 2).

### Characteristics of included studies

This review included 14 RCTs, 12 parallel, and two cross-over designs. The total population was 395 patients, most of whom were women, and their ages ranged from 21 to 69 years. Eleven RCTs compared BTX-A injections to saline placebo injections [7, 41, 42, 44–51]. A total of 207 patients received BTX injections, 116 received placebo saline injections, and 18 were used as control samples and did not receive any treatment. There were ten control patients in Zhang et al. [51] and 8 in Sewane et al. [49]. Al-Wayli [39], De Carli et al. [40], and Guarda-Nardini et al. [43] used active treatments as comparators. Al-Wayli [39] compared BTX-A treatment to conventional methods in the control group, which used behavioral strategies (n = 25). De Carli et al. [40] compared the effectiveness of BTX injection to low-energy laser therapy (n = 8). Guarda-Nardini et al. [43] used fascial manipulation (n = 15) as a comparator.

### Results of the included studies

**Pain reduction.**  In research investigations that examined pain as a primary measure, the Visual Analog Scale (VAS) was employed as the standard assessment tool, except for Kurtoglu et al. [45], who opted to administer the Research Diagnostic Criteria for Temporomandibular Disorders (RDC/TMD) Axis II Biobehavioral Questionnaire to assess pain intensity.

In Ernberg et al. [41], at a 1-month follow-up, BTX-A demonstrated a higher average percentage of pain reduction than saline. On a 0–100 VAS scale, the average pain reduction after BTX-A was 30 points; after saline, it was 11. At the 3-month follow-up, pain reduction was 23 and 4 points in the BTX-A and saline groups, respectively. Guarda-Nardini et al. [42] reported improvements in pain at rest from 5.00±3.62 at baseline to 3.60 ±2.88 for BTX vs from 3.90

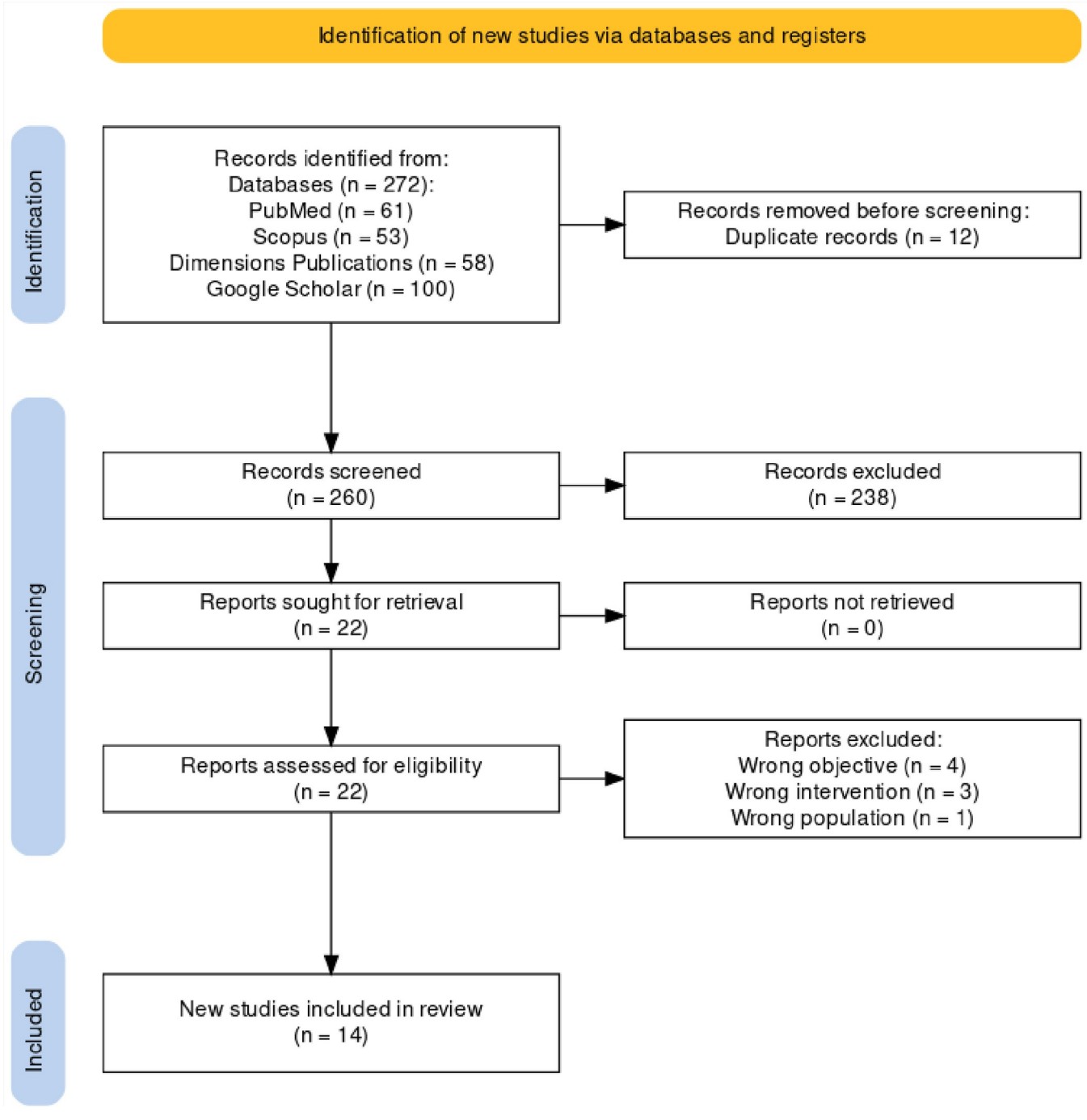

**Fig 1. PRISMA flowchart showing the study selection process.**

±2.92 at baseline to 4.10± 2.85 for placebo at six months. They also reported a reduction in pain during chewing from 6.20 ± 2.78 at baseline to 3.60 ± 2.37 for the BTX group vs. 4.10 ± 2.92 at baseline to 4.70± 2.79 for placebo at six months. This study observed that the Botox-treated group exhibited greater reductions in pain scores compared to the placebo-treated group at baseline, as well as at the 1-week, 1-month, and 6-month postoperative intervals. In a study conducted by Kim et al. [44], the authors observed an enhancement in pain

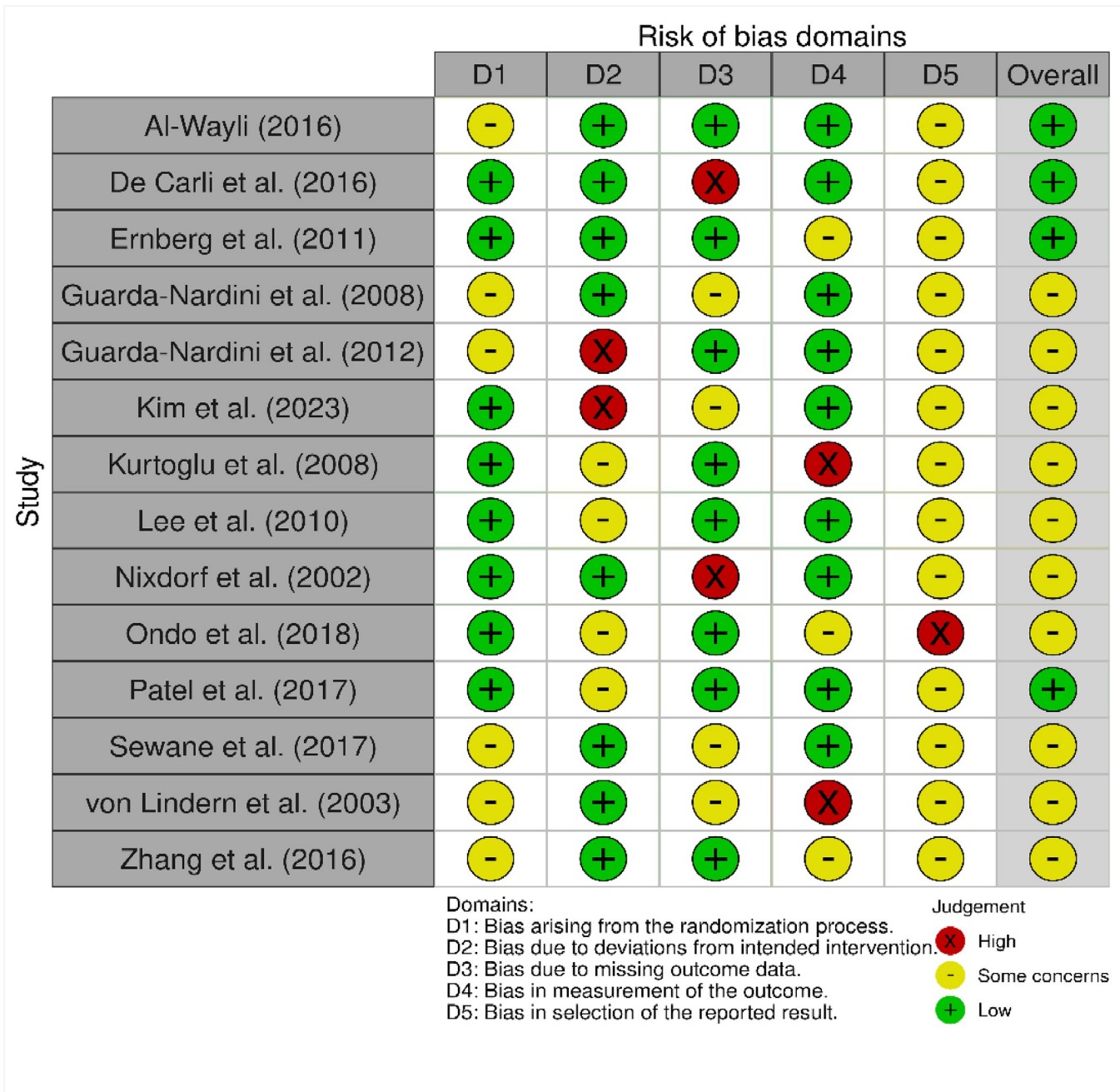

**Fig 2. Risk of bias graph.**

scores for the group receiving BTX compared to the group receiving a placebo, specifically in relation to orofacial pain intensity (OVAS), after a duration of four weeks. (2.64 ± 2.02 vs 3.64 ± 2.06), eight weeks (2.39 ± 2.24 vs. 3.79 ± 2.00) and 12 weeks (2.50 ± 2.31 vs. 3.43 ± 1.81). However, the changes in OVAS were not significantly different between the BTX and placebo groups. From the analysis by Ondo et al. [48], the change in VAS for pain (2-tailed t-test) was better in the onabotulinum toxin-A (BoNT-A) group. The mean pain scores in Patel et al. [7] for the IncobotulinumtoxinA group at the 4-week mark were significantly lower than those for

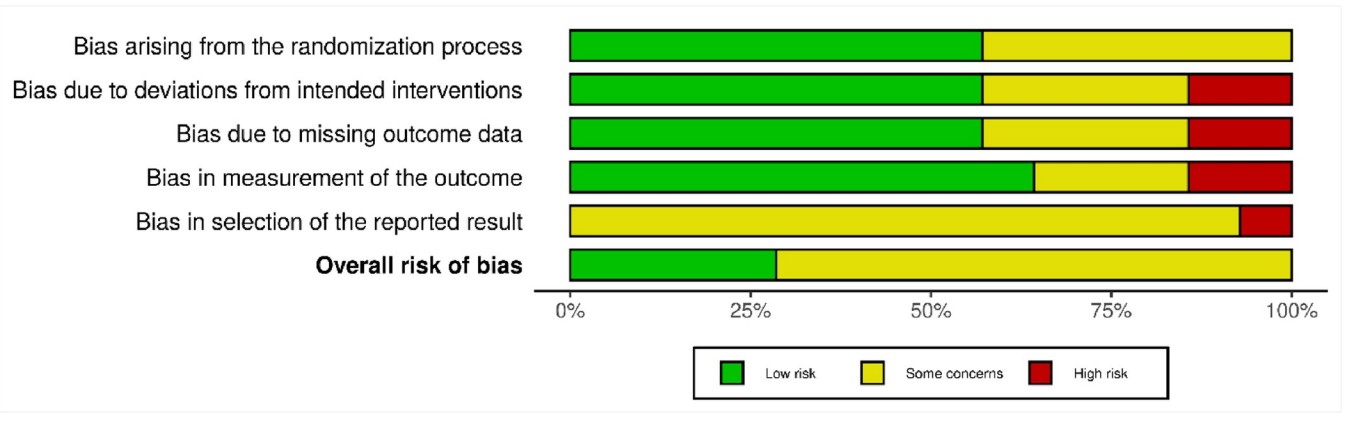

**Fig 3. Risk of bias summary.**

the placebo group (0.9 ± 1.7 vs 3.72 ± 2.5, P = 0.009). Mean pain scores decreased significantly between the groups (change in placebo = 1.7, P = .01 and change in IncobotulinumtoxinA = 4.5, P = .0002). The findings suggest that the cohort receiving ncobotulinumtoxinA experienced a notable decrease in pain levels compared to the cohort receiving a placebo. According to research conducted by Sewane et al. [49], the group receiving BTX-A experienced a reduction in pain levels both at rest and during chewing. In contrast, the placebo and control groups did not exhibit any significant changes in pain levels. The study by von Lindern et al. [50] found that 91% of BTX patients had a mean visual analog pain scale decrease of 3.2. This reduction differed remarkably from that in the placebo group (p < 0.01). Von Lindern et al. [50] also reported that patients experiencing higher levels of pain (VAS score $\geq$ 6.5) showed a substantial improvement ($\geq$ 3.5, total of 26 individuals), whereas those with lower pain levels (VAS score < 6.5) showed only modest improvement ($\leq$ 3.5, total of 27 individuals). In a study by Kurtoglu et al., BTX improved pain scores. [45]. A decrease in masseter muscle action potentials was observed on day 14, followed by an increase on day 28. Nevertheless, the decrease in pain scores persisted through day 28. Nixdorf et al. [47] found no statistically significant (P = 0.10) difference in pain intensity between BTX and placebo groups. In the BTX-A group, pain intensity decreased by 19 mm (SD = 31), while in the placebo group, it decreased by 1 mm (SD = 16).

In Al-Wayli [39], three weeks after treatment, the BTX-A group had a significantly lower mean pain score than the control group (which used traditional methods) (4.6 ± 0.58 vs. 5.4 ± 0.58). In the second and fifth months, the scores were 2.5 ± 0.59 vs 4.3 ± 0.48 and 0.2 ± 0.51 vs 2.1 ± 0.74, for the BTX-A and control group, respectively. In another placebo-free trial, De Carli et al. [40] found that the laser group had a statistically significant reduction in pain after 12 days of irradiation, but the BTX group took 30 days to see an improvement. However, there was no statistically significant difference between the two groups regarding pain symptom reduction 30 days into the trial (p = 0.985). In Guarda-Nardini et al. [43], VAS pain levels in the BTX group decreased to 5.2 ±2.1 immediately after the injection and to 4.8 ± 2.0 at the three-month follow-up. In contrast, the group receiving facial manipulation experienced values of 2.1 ± 1.4 immediately after the injection and 2.5 ± 2.2 at the three-month follow-up. Fascial manipulation was superior in reducing subjective pain perception, although these results were not statistically significant.

**Meta-analysis for pain reduction.** The results of Guarda-Nardini et al. [42], Patel et al. [7], Kim et al. [44], and Sewane et al. [49] were similar and sufficiently consistent to be pooled in a meta-analysis (Fig 4). One month after treatment, the outcome of pain reduction favored

**Table 2. Study descriptor table.**

| Study | Country | RCT Design | Participants | Type of TMD | Diagnoses method | Intervention group details | Comparator details | Studies outcomes | Follow-up period |
|---|---|---|---|---|---|---|---|---|---|
| Al-Wayli [39] | Saudi Arabia | Parallell | N = 50, 100% women, mean age of 45.5 ± 10.8 yrs | Masseter muscle pain and bruxism | ICSD-2 | N = 25, 20 U each side • Masseter (3 points) | Conventional treatment (N = 25), Behavioral strategies, occlusal splints, and pharmacologic measures | Pain (VAS), Adverse events | 2, 6 and 12 months |
| De Carli et al. [40] | Brazil | Parallel | N = 15, 86.6% women, mean age of 30 yrs | Myofascial pain (> 1 mo), bruxism, clenching or tooth wear | clinical examination | N = 7, 100 U on each side • Masseter (2 points of 30 U) • Temporalis (1 point of 20 U) After 15 d: • Masseter (2 points of 30 U) • Temporalis (1 point of 15 U) | Low-energy laser therapy (N = 8) | Pain (VAS), Mouth opening (in mm) | 1 month |
| Ernberg et al. [41] | Sweden and Denmark | Cross-over | N = 21, 90% women, mean age of 38 ± 12 yrs | TMD | DC/TMD | N = 12, 100 U on each side • Masseter (2 points of 50) | Placebo (N = 9) Saline injections | Pain (VAS), Adverse events | 1 and 3 month |
| Guarda-Nardini et al. [42] | Italy | Parallel | N = 20, 50% women, mean age of 38 ± 12 yrs | Bruxism and myofascial pain | DC/TMD | N = 10, 100U each side • Masseter (2 points of 30 U) • Temporalis (3 points of 20 U) | Placebo (N = 10) Saline injections | Pain (VAS), Mouth opening (in mm), Adverse events | 1 and 3 month |
| Guarda-Nardini et al. [43] | Italy | Parallel | N = 30, 73.3% women, Age range: 26–69 y | TMD | DC/TMD | N = 15, 150 U on each side • Masseter and temporalis (mean of 5 points) | Fascial manipulation (N = 15) Deep digital pressure | Pain (VAS), Mouth opening (in mm), Adverse events | 3 months |
| Kim et al. [44] | South Korea | Parallel | N = 21, 90% women, the age range of 21–53 yrs, mean age of 33.95 ± 9.70 yrs | Myogenous TMD | DC/TMD | N = 14, 150 U on both sides | Placebo, (n = 7) saline injections | Pain (VAS), Adverse events | 1 2, and 3 month |
| Kurtoglu et al. [45] | Turkey | Parallel | N = 24, 79.1% women, mean age of 29.6 ± 12.7 (BTX group)/ 23.4 ± 4.7 (Placebo group) | TMD | DC/TMD | N = 12, 100 U on each side • Masseter (3 points of 10 U) •Temporalis (2 points of 10 U) | Placebo (N = 12), Saline injections | Pain (RDC/ TMD Axis II), Adverse events | 1 month |
| Lee et al. [46] | South Korea | Parallel | n = 12, 42% women, mean age of 25.0 ± 2.35 yrs for men and 24.8 ± 0.83 yrs for women | Nocturnal Bruxism | Clinical examination | N = 6, 80 U on each side | Placebo (N = 6), Saline injections | Bruxism events | 1 2, and 3 month |
| Nixdorf et al. [47] | Canada | Cross-over | N = 10, 100% women, mean age of 33 yrs | TMD | DC/TMD | N = 5, 75 U on each side • 50 U masseter (3 points) •25 U temporal (3 points) | Placebo (N = 5), Saline injections | Pain (VAS), Mouth opening (in mm), Adverse events | 2 and 4 months |

*(Continued)*

**Table 2.** (Continued)

| Study | Country | RCT Design | Participants | Type of TMD | Diagnoses method | Intervention group details | Comparator details | Studies outcomes | Follow-up period |
|---|---|---|---|---|---|---|---|---|---|
| Ondo et al. [48] | USA | Parallel | N = 23, 82.6% women, mean age 47.4 ± 16.9 yrs | Bruxism | ICSD-3; EMG of the masseter and temporalis muscles | N = 13, 100 U (each side): • 30 U masseter (2 points) • 20 U temporalis (3 points) | Placebo (N = 10), Saline injections | Pain (VAS), Adverse events | 1 month |
| Patel et al. [7] | USA | Parallel | N = 20 | TMD | pain >3 on a 0–10 ordinal scale; at least 10 d per month | N = 10, 85 U (each side): • 50 U masseter • 25 U temporalis • 10 U external pterygoid | Placebo (N = 9), Saline injections | Pain (VAS), Adverse events | 1 month |
| Sewane et al. [49] | India | Parallel | N = 24 | Bruxism and myofascial pain of masticatory muscles | >5 episodes/week, grinding sounds during the morning, masticatory muscle or pain. | N = 8, 100 U on each side • Masseter (2 points of 30 U) • Temporalis (2 points of 20 U) | Placebo (N = 8) Saline injections: Control (N = 8) No treatment | Pain (VAS) | 3 and 6 months |
| von Lindern et al. [50] | Germany | Parallel | N = 90 | Facial pain due to masticatory muscle hyperactivity | clinical examination (MFP with bruxism) | N = 60, 35 U on each side • Masseter, temporalis, and pterygoideus medialis muscles. | Placebo (N = 30), Saline injections | Pain (VAS), Adverse events | 1 and 3 months |
| Zhang et al. [51] | China | Parallel | N = 30, 13% women, age range of 25–37 y | TMD and bruxism | clinical examination | N = 10, 100 U each side • Masseter (3 points) | Placebo (N = 10) Saline injections; Control (N = 10) No treatment | Adverse events | 6 months |

DC/TMD—Diagnostic Criteria for Temporomandibular Disorders [32], ICSD - International Classification of Sleep Disorder, EMG– electromyography.

the BTX-A group, with Mean Difference (MD) = −1.71 (95% CI, −2.87 to −0.5), p = 0.37, and $I^2$ = 0%. In addition, at the 3- and 6-month follow-ups, the mean difference still favored BTX-A treatment. At three months, MD = −1.53 (95% CI, −2.80 to −0.27), p = 0.15 and $I^2$ = 47%, and at six months, the MD = −1.33 (95% CI, −2.74 to 0.77), p = 0.15 and $I^2$ = 51%. Nevertheless, the statistical analysis revealed that there were no significant mean differences between the groups receiving BTX-A and placebo across all timeframes. This indicates that the administration of BTX-A did not yield superior outcomes compared to placebo in terms of pain reduction among patients with temporomandibular disorder (TMD) at 1, 3, and 6 months.

Ernberg et al. [41] and Ondo et al. [48] used a 0–100 VAS scale to examine pain. However, their data could not be pooled in the meta-analysis because, unlike other studies, Ernberg et al. [41] and Ondo et al. [48] reported VAS scores in terms of change instead of mean group scores.

**Any adverse events.** Ernberg et al. [41] reported 9/12 BTX adverse events and 5/9 placebo incidents. The most frequent side effects in the study were headache, tiredness resembling muscle weakness, jaw pain, influenza-like symptoms, and dry mouth. In Ondo et al. [48], the only reported side effects were in the BTX-A group, where two patients complained of cosmetic alteration in their smile. No adverse events were reported by Patel et al. [7], Guarda-Nardini et al. [43], and Lee et al. [46].

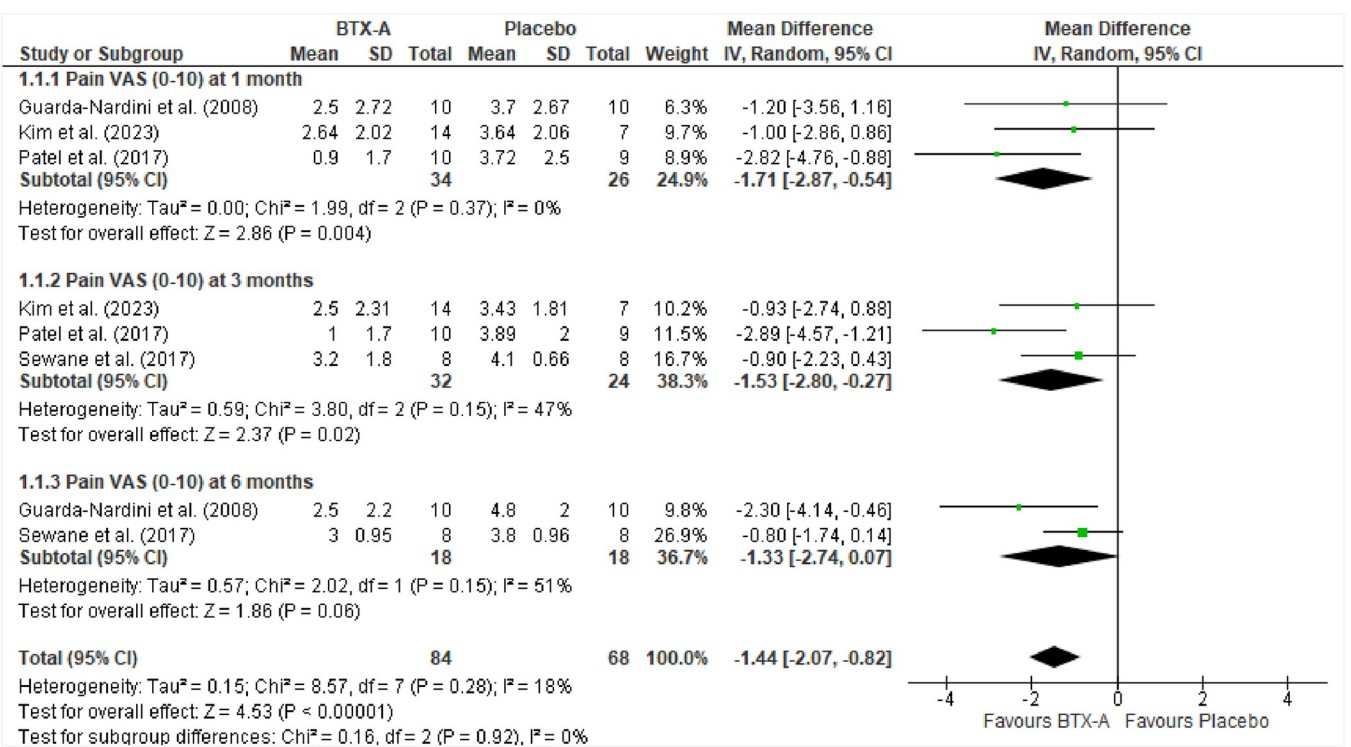

**Fig 4. Forest plot for pain scores on VAS (0–10).**

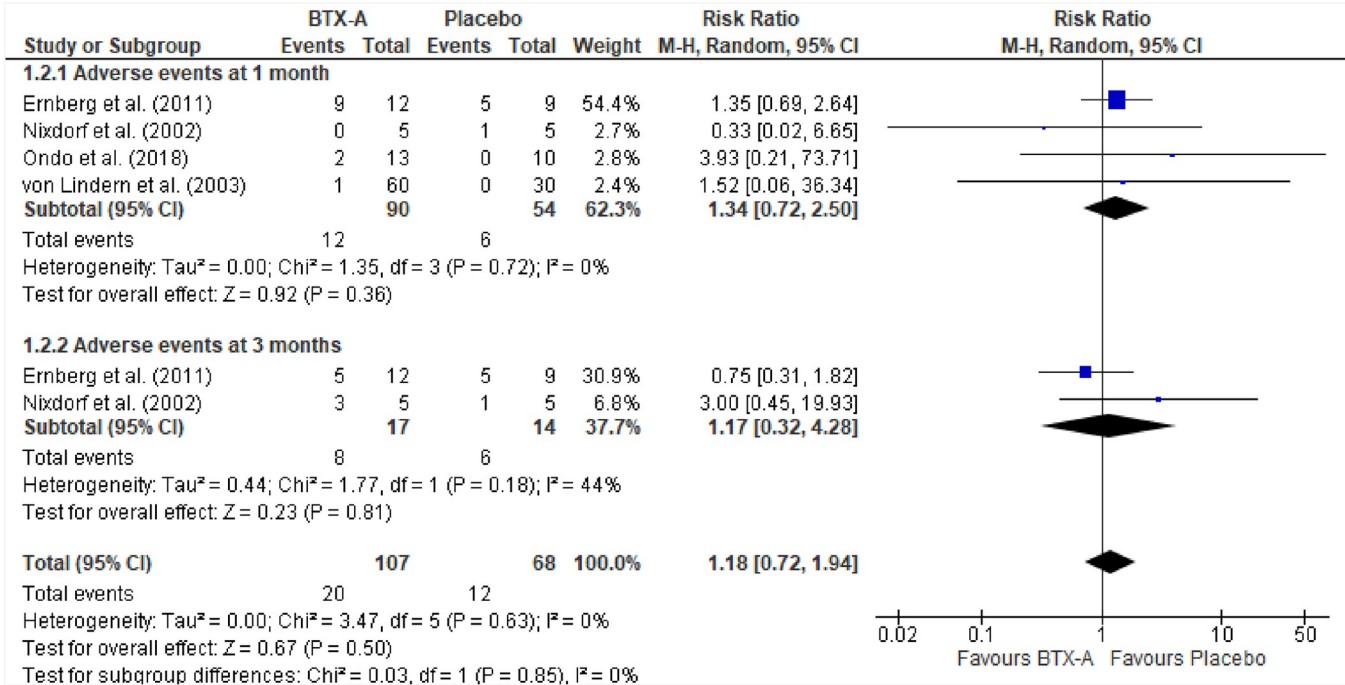

**Fig 5. The risk of adverse events.**

**Meta-analysis on adverse events.** Data from Ernberg et al. [41], Nixdorf et al. [47], Ondo et al. [48], and von Lindern et al. [50] were used to calculate the risk of side effects. The results (Fig 5) showed that the placebo group had an increased risk of RR = 1.34 (95%CI, 0.48–6.78), p = 0.71 at one month, and RR = 1.17 (95%CI, 0.54–3.88), p = 0.18 at three months, for developing side effects. However, this effect was not statistically significant.

**Maximum mouth opening.** De Carli et al. [40] reported that neither the BTX treatment nor the low-energy laser group (comparator) exhibited a notable increase in mouth opening during treatment. In the study by Guarda-Nardini et al. [42], there were no significant differences in the range of mouth opening, assisted and non-assisted. The alterations in maximum mouth opening, measured in millimeters (mm), in a study conducted by Kim et al. [44] did not yield statistically significant differences between the groups that received BTX and the placebo group. The BTX group in Guarda-Nardini et al. [43] demonstrated improvement in mouth opening at an average of 51.4 mm, which was lower compared to 52.4 mm in the group receiving fascial manipulation. In Nixdorf et al. [47], patients receiving BTX-A demonstrated a significantly narrower maximum opening (with and without pain) than the placebo group.

**Bruxism events.** The number of EMG bruxism events in the BTX group in Lee et al. [46] decreased following the injection of botulinum toxin into the masseter muscle but not in the temporalis muscle. This decrease was significantly different post-injection but did not differ in the subsequent three follow-ups. Ondo et al. [48] reported a significant variability in nightly bruxism events. In the BoNT group, the overall number of events decreased from 9.18 ± 8.48/h to 6.95 ± 7.04/h. In contrast, the placebo group showed an increase in events from 4.63 ± 3.45/h to 10.65 ± 9.57/h (p = 0.09, two-tailed t-test for change).

**Maximum occlusal force.** Sewane et al. [49] and Zhang et al. [51] found that the maximal occlusal force was significantly different in the BTX-A group compared to the placebo and control groups. There was no statistical difference between the placebo and control groups in either study. The maximal occlusal force in the BTX-A group changed significantly with time, with the lowest value observed three months following treatment, according to Sewane et al. [49]. The value was shown to be statistically insignificantly lower six months after therapy compared to the pre-treatment value.

## Discussion

The purpose of this study was to assess the effectiveness of BTX-A in the treatment of TMD, with pain reduction as the primary outcome and adverse events, maximum mouth opening, bruxism, and maximum occlusal force as secondary outcomes. Several studies have reported positive results in pain reduction with BTX-A treatment compared to placebo [41, 42, 44, 48]. However, the statistical analysis showed no statistically significant differences between the BTX and placebo groups regarding pain reduction at 1, 3, and 6 months after treatment. The study also revealed that, compared to the placebo group, the BTX group showed no better results for adverse events at 1 and 3 months post-treatment. BTX-A was reported to be more efficacious in pain reduction than conventional treatments such as behavioral intervention, occlusal splints, or medication at 1, 3, and 12 months [39]. Additionally, it proved to be more effective than low-level laser therapy after one month. [40]. Guarda-Nardini et al. [43] reported that BTX-A was both statistically and clinically less efficacious than facial manipulation in reducing pain three months after the treatment. Other outcomes have also been reported. The use of BTX did not significantly improve mouth opening, maximum occlusal force, or decrease bruxism events compared to placebo and other treatments [40, 42–44, 46–49, 51]. These findings show that BTX injections offer no significantly better outcomes than placebo and other treatments for pain reduction, adverse events, mouth opening, maximum occlusal force, or decrease in bruxism events.

The findings of this review do not support the clinical use of BTX injections for managing temporomandibular disorders. BTX offers no better results than no treatment or other treatments. Even though individual studies reported the positive impact of BTX, these results did not suffice when pooled together and subjected to statistical analysis. However, these conclusions are not robust due to the quality of evidence included in this review. The quality of the evidence was low to moderate due to a moderate to high risk of bias among the studies. This review calls for carrying out more rigorous RCTS with higher quality detailing of methodology and results. There was also high study variability in the assessed outcomes. Despite using the VAS for pain, two studies used the 0–100 scale instead of the 0–10 scale, which was most commonly used. In addition, the two studies that used the 0–100 VAS scale were incompatible for pooling in a meta-analysis due to how the pain scores were reported.

For future research, it is recommended to extend follow-up periods to adequately capture the sustained effects of the intervention. Considering that BTX is active for up to six months, investigations with longer follow-up durations would be beneficial. However, only Sewane et al. [49] and Zhang et al. [51] had a 6-month follow-up, with Al-Wayli [39] having the most extended period of 12 months. In addition, given the variability among patients regarding optimal dosing frequency and duration of effect, incorporating patient-directed treatment strategies is crucial for future trials, ensuring a more comprehensive evaluation over an appropriate follow-up duration.

Shimada et al.'s study [52] focussed on pain intensity and range of movements as outcome parameters. Mobilization exercise, including manual therapy, passive jaw mobilization with oral appliances, and voluntary jaw exercise, appeared to be a promising option for painful TMD conditions such as myalgia and arthralgia.

Schiffman et al. [53], in their research, recommended Diagnostic Criteria for TMD (DC/TMD). In their study, the Axis I protocol included both a valid screener for detecting any pain-related TMD and valid diagnostic criteria for differentiating the most common pain-related TMD. Their recommended evidence-based new DC/TMD protocol is appropriate for use in both clinical and research settings. These validated instruments allow for the identification of patients with a range of simple to complex TMD presentations.

In their study, Ferrilli et al. [54] assessed the linkage between hypovitaminosis D and TMDs. Results of their study showed that vitamin D serum levels could often be lower in patients with a TMD. Data from their research also suggested that vitamin D serum levels and VDRs might have a role in the onset and progression of TMDs.

Gil-Martinez A. et al. [55], in their research, evaluated the evidence, identified challenges, and proposed solutions from a clinical point of view for patients with craniofacial pain and TMD. They propose a new biobehavioral model regarding pain perception and motor behavior for the diagnosis and treatment of patients with painful TMD. Their results suggest that long-term treatment is needed to obtain the maximal effects of all these drugs, which sometimes become evident only after several weeks of treatment. Regarding Botulinium, there were mixed results; of the five studies included, two obtained a significant reduction in pain, one showed equal effects compared with masticatory manual therapy, and two showed no significant differences for BTX compared with placebo. They proposed a multidisciplinary way for the treatment of TMD, including the therapeutic interventions of physiotherapists, dentists, psychologists, and physicians.

## Conclusion

Several studies independently reported positive outcomes for BXT compared to placebo [41], [42, 44, 48]. A meta-analysis showed that BTX was not better than placebo in terms of pain

reduction and adverse events. However, study reports showed that BTX was better than conventional treatment and low-energy laser therapy [39, 40] and less effective than fascial manipulation for pain reduction [43]. The use of BTX did not significantly improve mouth opening, maximum occlusal force, or decrease bruxism events compared to placebo and other treatments [40, 42–44, 46–49, 51]. The findings in this review show that BTX injections offer no better outcomes for TMD management than placebo and other treatments.

## Supporting information

**S1 Checklist. Prisma checklist for reporting systematic reviews.**
(DOCX)

**S1 File. Supporting information including the data extraction Excel file, the quality assessment scoring sheet, and the Review Manager software file.**
(ZIP)

## Author Contributions

**Conceptualization:** Ravinder S. Saini, Vishwanath Gurumurthy.

**Data curation:** Muhammad Ali Abdullah Almoyad, Shashit Shetty Bavabeedu.

**Formal analysis:** Muhammad Ali Abdullah Almoyad, Shashit Shetty Bavabeedu.

**Investigation:** Syed Altafuddin Quadri, Mohammed Saheer Kuruniyan, Punnoth Poonkuzhi Naseef.

**Methodology:** Ravinder S. Saini, Vishwanath Gurumurthy.

**Project administration:** Ravinder S. Saini, Artak Heboyan.

**Resources:** Syed Altafuddin Quadri, Mohammed Saheer Kuruniyan.

**Supervision:** Ravinder S. Saini, Artak Heboyan.

**Writing – original draft:** Ravinder S. Saini, Rayan Ibrahim H. Binduhayyim, Artak Heboyan.

**Writing – review & editing:** Shashit Shetty Bavabeedu, Mohammed Saheer Kuruniyan, Seyed Ali Mosaddad, Artak Heboyan.

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
