## [Decision Letter · Decision Letter 0]

20 Dec 2023

PONE-D-23-40027The Effectiveness of Botulinum Toxin for Temporomandibular Disorders: A Systematic Review and Meta-analysisPLOS ONE

Dear Dr. Mosaddad,

Thank you for submitting your manuscript to PLOS ONE. After careful consideration, we feel that it has merit but does not fully meet PLOS ONE’s publication criteria as it currently stands. Therefore, we invite you to submit a revised version of the manuscript that addresses the points raised during the review process.

We look forward to receiving your revised manuscript.

Kind regards,

Martina Ferrillo

Academic Editor

PLOS ONE

Journal Requirements:

   "All the authors are thankfull to the King Khalid University, Saudi Arabia for the financial Support"

    "Deanship of Scientific Research at King Khalid University partially funded this work through Large Group RGP 2/348/44."

   "Deanship of Scientific Research at King Khalid University partially funded this work through Large Group RGP 2/348/44."

5. We note that this manuscript is a systematic review or meta-analysis; our author guidelines therefore require that you use PRISMA guidance to help improve reporting quality of this type of study. Please upload copies of the completed PRISMA checklist as Supporting Information with a file name “PRISMA checklist”.

Additional Editor Comments:

Please, modify the paper according to they suggestions. The paper will be reconsidered after revision.

Reviewers' comments:

Reviewer's Responses to Questions

**Comments to the Author**

1. Is the manuscript technically sound, and do the data support the conclusions?

Reviewer #1: Yes

Reviewer #2: Partly

2. Has the statistical analysis been performed appropriately and rigorously? 

Reviewer #1: Yes

Reviewer #2: N/A

3. Have the authors made all data underlying the findings in their manuscript fully available?

Reviewer #1: Yes

Reviewer #2: Yes

4. Is the manuscript presented in an intelligible fashion and written in standard English?

Reviewer #1: No

Reviewer #2: No

5. Review Comments to the Author

Reviewer #1: Dear Authors,

I have read your paper with great interest. This systematic review with metanalysis aims to examine the efficacy of BTX in treating TMD. The topis is actual and innovative, and in line with the editorial goal of PLOS. Nevertheless, some critical issue should be addressed:

Introduction: The introduction is lack of information regarding the possible mini-invasive treatment of TMD (e.g ozone-therapy, hyaluronic acid etc.). Please, read doi: 10.1111/joor.13571.

Material and Methods: The authors have included in PICO model partecipants suffered TMD. How was the diagnosis performed? this point is crucial to better clarify the TMD . I suggest to read PMID: 20213030.

Moreover, I suggest to add in table a brief protocol strategy for BTX for each study with the muscle involved in the protocol therapy

Results: well done

Discussion: The discussion is a simple list of the results of the research. I suggest to rewrite it at light of the potential use of BTX in muscular and neuromuscular disease.

Please, move the paragraph "Botulinum toxin's possible action pathways in treating TMD" in the introduction.

Best Regards

Reviewer #2: Dear Authors,

The topic is very interesting, and the work is well described, although some major issues should be addressed. In general, the work seems difficult to read and should benefit of a language revision.

Abstract

In my opinion the abstract should be revised in order to better describe the object of this Systematic Review, also describing controls and outcomes. Keywords utilization is not essential, as for headings, as several recent works in this Journal have an unstructured abstract

Introduction

- The reference should be placed in square brackets.

- In this section I suggest also to describe the various treatment possibility, including physical therapy and medications, other than splint.

- It is also necessary to underpin the muscle treated with botulinum toxin across the studies

- Lastly the objective should be revised in order to better include population, study selection, outcomes and controls, and should be written in scientific language (“The goal of this study is to determine if botulinum toxin can treat TMJ problems” seems too much colloquial for a systematic review with metanalysis)

- Botulinum toxin's possible action pathways in treating TMD should be describe also in this section (other than in discussion)

- Other medical condition related to TMD should be evaluated in this section

You might consider reading these recent works:

• Shimada A, Ishigaki S, Matsuka Y, Komiyama O, Torisu T, Oono Y, Sato H, Naganawa T, Mine A, Yamazaki Y, Okura K, Sakuma Y, Sasaki K. Effects of exercise therapy on painful temporomandibular disorders. J Oral Rehabil. 2019 May;46(5):475-481. doi: 10.1111/joor.12770. Epub 2019 Feb 19. PMID: 30664815.

• Schiffman E, Ohrbach R, Truelove E, Look J, Anderson G, Goulet JP, List T, Svensson P, Gonzalez Y, Lobbezoo F, Michelotti A, Brooks SL, Ceusters W, Drangsholt M, Ettlin D, Gaul C, Goldberg LJ, Haythornthwaite JA, Hollender L, Jensen R, John MT, De Laat A, de Leeuw R, Maixner W, van der Meulen M, Murray GM, Nixdorf DR, Palla S, Petersson A, Pionchon P, Smith B, Visscher CM, Zakrzewska J, Dworkin SF; International RDC/TMD Consortium Network, International association for Dental Research; Orofacial Pain Special Interest Group, International Association for the Study of Pain. Diagnostic Criteria for Temporomandibular Disorders (DC/TMD) for Clinical and Research Applications: recommendations of the International RDC/TMD Consortium Network* and Orofacial Pain Special Interest Group†. J Oral Facial Pain Headache. 2014 Winter;28(1):6-27. doi: 10.11607/jop.1151. PMID: 24482784; PMCID: PMC4478082.

• Ferrillo M, Lippi L, Giudice A, Calafiore D, Paolucci T, Renò F, Migliario M, Fortunato L, Invernizzi M, de Sire A. Temporomandibular Disorders and Vitamin D Deficiency: What Is the Linkage between These Conditions? A Systematic Review. J Clin Med. 2022 Oct 22;11(21):6231. doi: 10.3390/jcm11216231. PMID: 36362456; PMCID: PMC9655046.

• Gil-Martínez A, Paris-Alemany A, López-de-Uralde-Villanueva I, La Touche R. Management of pain in patients with temporomandibular disorder (TMD): challenges and solutions. J Pain Res. 2018 Mar 16;11:571-587. doi: 10.2147/JPR.S127950. PMID: 29588615; PMCID: PMC5859913.

Materials and Methods

-In this section i suggest to separate the primary outcome and the preferred scale evaluation (VAS), from the secondary outcomes.

- Was the string utilized also on “Dimension Publication”?

- “Clinical trials without a population could only be eligible if the results for the adult population were reported separately” this section should be revised to better explain the study selection

- “The trial's primary objective had to be examining the effectiveness of BTX in managing TMD” the objective of the systematic review seems to be the effectiveness of BTX in treating pain in patients with TMD, as the effectiveness is not a quantifiable outcome and authors utilized VAS score

Results

- In the subheading describing the population, the ratio male

/female and other discrete variables should be explicated as percentages, while continuous variables should be written as mean and standard deviation

- in this section I suggest to better compare the various control treatments between the studies

- The metanalysis for VAS only include 2 to 3 studies at time, as it might be inconclusive for this type of study. I have also doubts regarding the adverse events analysis, where RR was 1,34 times higher for placebo group at one month compared with botulinum toxin

- author stated that “Nevertheless, the statistical analysis revealed that there were no significant mean differences seen between the groups receiving BTX-A and placebo across all timeframes. This indicates that the administration of BTX-A did not yield superior outcomes compared to the absence of treatment in terms of pain reduction among patients with temporomandibular disorder (TMD) at 1, 3, and 6 months” this section is in contrast with all the paper until that, and should be better explained.

Discussion

- Study limitation should be implemented according to this revision

- In Author’s contribution name and surname should be pointed

- Lastly, references are only partly formatted in accordance with this Journal instruction for Authors.

6. PLOS authors have the option to publish the peer review history of their article (what does this mean?). If published, this will include your full peer review and any attached files.

Reviewer #1: No

Reviewer #2: No

---

## [Author Response · Author response to Decision Letter 0]

4 Feb 2024

A response letter has been uploaded, addressing all comments from reviewers.

---

## [Decision Letter · Decision Letter 1]

16 Feb 2024

The Effectiveness of Botulinum Toxin for Temporomandibular Disorders: A Systematic Review and Meta-analysis

PONE-D-23-40027R1

Dear Dr. Seyed Ali Mosaddad,

We’re pleased to inform you that your manuscript has been judged scientifically suitable for publication and will be formally accepted for publication once it meets all outstanding technical requirements.

Kind regards,

Martina Ferrillo

Academic Editor

PLOS ONE

Additional Editor Comments (optional):

Based on reviewers' comments, the paper is suitable for publication

Reviewers' comments:

Reviewer's Responses to Questions

**Comments to the Author**

1. If the authors have adequately addressed your comments raised in a previous round of review and you feel that this manuscript is now acceptable for publication, you may indicate that here to bypass the “Comments to the Author” section, enter your conflict of interest statement in the “Confidential to Editor” section, and submit your "Accept" recommendation.

Reviewer #1: All comments have been addressed

Reviewer #2: All comments have been addressed

2. Is the manuscript technically sound, and do the data support the conclusions?

Reviewer #1: Yes

Reviewer #2: Yes

3. Has the statistical analysis been performed appropriately and rigorously? 

Reviewer #1: Yes

Reviewer #2: Yes

4. Have the authors made all data underlying the findings in their manuscript fully available?

Reviewer #1: Yes

Reviewer #2: Yes

5. Is the manuscript presented in an intelligible fashion and written in standard English?

Reviewer #1: Yes

Reviewer #2: Yes

6. Review Comments to the Author

Reviewer #1: Dear Authors,

all my concerns have been addressed. Please, format the references following the editorial line of Plos one.

Best

Reviewer #2: The manuscript has improved substantially and most of the reviewers' questions and concerns have been addressed

7. PLOS authors have the option to publish the peer review history of their article (what does this mean?). If published, this will include your full peer review and any attached files.

Reviewer #1: No

Reviewer #2: No

---

## [Editor Report · Acceptance letter]

1 Mar 2024

PONE-D-23-40027R1 

PLOS ONE

Dear Dr. Mosaddad, 

I'm pleased to inform you that your manuscript has been deemed suitable for publication in PLOS ONE. Congratulations! Your manuscript is now being handed over to our production team.

Kind regards, 

on behalf of

Dr. Martina Ferrillo 

Academic Editor

PLOS ONE